# “*We’ve Always Been Kind of Kicked to the Curb*”: A Mixed-Methods Assessment of Discrimination Experiences among College Students

**DOI:** 10.3390/ijerph19159607

**Published:** 2022-08-04

**Authors:** Cindy Mahoney, Benjamin J. Becerra, Devin Arias, Jacqueline E. Romano, Monideepa B. Becerra

**Affiliations:** 1Department of Health Science and Human Ecology, Center for Health Equity, California State University, San Bernardino, CA 92407, USA; 2Department of Information and Decision Sciences, Center for Health Equity, California State University, San Bernardino, CA 92407, USA; 3Department of Teacher Education and Foundations, Center for Health Equity, California State University, San Bernardino, CA 92407, USA

**Keywords:** racism, colorism, elitism, linguistic racism, millennials, ageism, sexism

## Abstract

**Background:** Experiences of discrimination are prevalent among minority populations, although often empirical evidence does not provide depth into the source and types of discrimination, such as racial/ethnic, gender-based, age, etc. The goal of this study was to assess the unique patterns, types, and sources of discrimination experiences that college students face and explore the role these experiences play in their mental health. **Methods:** An explanatory sequential mixed-methods study was utilized. Quantitative assessment of college students from a Hispanic and minority-serving institution was conducted to evaluate experiences of discrimination and its association to physical health and mental health (including psychological distress), as well as food insecurity, a marker for poverty. Next, qualitative data were thematically analyzed to further provide an in depth understanding on the sources of such experiences, types of discriminations, as well as the impact on mental health. **Results:** Results of the quantitative assessment highlight that discrimination was prevalent among the population with a higher everyday discrimination score significantly associated with serious psychological distress, low mental health status, low physical health status, and being food insecure. Further, most of the participants reported that they felt discriminated due to their appearance, with race/ethnicity and skin color as next most commonly cited reasons. Qualitative assessment further demonstrates distinct types of discrimination experiences from a variety of sources. Within a family, colorism and having an American accent while speaking a native language was a predominant source, while among peers, having a non-American accent was a primary source of discrimination experiences. Such experiences based on elitism, gender, and age (being younger) from the workplace were prevalent among the target population. Finally, feelings of isolation, not belonging, as well as negative impact on self-efficacy and self-worth were noted. **Conclusion:** Experiences of discrimination are prevalent among college students, including from within family and peers. To improve mental health outcomes of such a population, campus-based measures are needed to promote resiliency and social support, as well as community-based initiatives to promote workplace training to create inclusive environments for younger generations entering the workforce.

## 1. Background

College students face a disproportionate share of health disparities, including poor sleep health [1,2,3,4], risky financial behaviors, such as credit card debt, health behaviors, and [5,6,7] low health literacy [8], including low sexual and reproductive health literacy [9,10,11,12], higher basic needs insecurities [13,14], as well as poor mental health outcomes [15,16,17,18]. For example, in an assessment of over 7600 college students from six different universities, Becker et al. noted that a majority (62%) of participants met the criteria for poor sleep, and such outcomes were associated with poor mental health status as well [1]. Likewise, a review of existing literature on HIV risk behaviors among college students noted that not only did a majority of participants have multiple sex partners, both safe sex communication and practices were also limited [7]. Further, in an assessment of basic need insecurities among college students, researchers have found that being food, finance, and housing insecurity were cumulatively related to increased odds of anxiety, depression, lower health status, as well as poor academic performance [13].

Another key emergent area of disparity noted among college students is that of discrimination. The American Psychological Association notes that discrimination is an “unfair or prejudicial treatment” of population based on shared characteristics, such as race, gender, age, sexual orientation, etc. [19]. Although prevalence studies on the various types and sources of discrimination among college students population remain limited, the empirical evidence notes that among African American college students, experiences of discrimination were associated with sleep problems [20]. Likewise, the literature notes that Hispanic/Latino college students with experiences of discriminations are more likely to develop symptoms of posttraumatic stress, as well as risky behaviors, such as alcohol use [21]. In addition, an assessment of female graduate students in science programs have also highlighted experiences of sexism, sexual harassment, gender role-based stereotyping, as well as microaggressions [22]. Although such literature highlights the prevalence and the burden of discrimination among college students, most are often limited to assessment of experiences of discrimination based on racial/ethnic identity and/or colorism and sometimes gender norms; delineation of types and sources of such discrimination remain limited as well. For example, recent evidence highlights the putative emergence of workplace discrimination among younger populations [23]. Likewise, studies among international students have noted a differing perception of discrimination based on European immigrant versus non-European [24]. Understanding the types of such experiences college students face, the source of such stressors, and how that, in turn, the impacts on mental health is imperative due to the critical transition phase of adapting to new environment, forging friendships, developing coping (or maladaptive coping) skills, etc. [25,26]. As such, the goal of this study was to assess (1) the prevalence of discrimination experiences, (2) association between such experiences with both physical and mental health, and (3) types and sources of such discrimination experiences among college students at a Hispanic and minority serving institution.

## 2. Methods

This study used an explanatory sequential mixed-methods approach, with quantitative assessment followed by that of qualitative. In this approach, qualitative data are utilized to explain, interpret, and provide further clarification of the quantitative results [27]. Combining the two methods in this sequence further allowed us to explain the patterns and associations noted in survey responses and, in turn, provide unique insight into emergent themes related to discrimination experiences among college students.

In this study, during the first phase, quantitative analysis of an annual student health assessment was conducted. Students, aged 18 years or older, were recruited from general education courses to ensure inclusion of a variety of majors and extra credit was provided as incentives. The annual survey collects data on various health and behavioral outcomes, including alcohol and tobacco use, sleep health, mental health, experiences of discrimination, and sociodemographic characteristics. In this study, we selected participants who identified as racial/ethnic minorities (chosen from a list provided). In the U.S., ethnicity is primarily defined as Hispanic/non-Hispanic, while racial groups may include (but not limited to) one or more of the following: African American, Asian American, White, etc. All participants who were currently enrolled, were at least 18 years of age, whose classes shared the survey, and who provided written consent were selected in the study. We did not collect any parental or family data. However, a majority of the participants at the institution are first-generation college students on financial aid. We further evaluated experiences of discrimination, assessed through the Everyday Discrimination scale [28], as well as the presence of psychological distress, evaluated using the Kessler-6 scale [29]. In addition, food security status, a marker for poverty, was assessed using the U.S. Department of Agriculture six-item questionnaire [30]. Finally, self-reported general mental and general physical health statuses were also included, similar to questions in the California Health Interview Survey, in addition to age, sex, and racial/ethnic identity. Due to most of the study population being Hispanic and, thus, low frequency of other racial/ethnic groups, we dichotomized the race/ethnicity variable by ethnicity only (Hispanic/Latino vs. not) to ensure protection of unique data combinations, per our ethical review board guidelines. A total of 308 participants were included for the quantitative assessment.

All quantitative data were analyzed in SPSS version 28 (IBM, Corp., Armonk, NY, USA). First, descriptive statistics were conducted to assess the mean discrimination score, as well as prevalence of psychological distress, low mental health status, and low physical health status. Next, bivariate analyses (independent sample *t*-tests) with alpha = 0.05 were used to assess whether mean discrimination differed by serious psychological distress status, mental health status, physical health status, and sociodemographic characteristics.

To help provide insight into the various types and sources of discrimination, we conducted qualitative analysis. For this second phase, students aged 18 years or older from five general education laboratories were recruited to ensure a diversity of majors. Those who consented to participate were given extra credit as an incentive. Further, due to the sensitive nature of the questions asked in shared spaces via focus groups, we did not analyze demographic characteristics, but used purposive sampling to ensure the sample size was reflective of the student population from the quantitative phase. Given that a majority of the students are considered vulnerable population (first generation, racial/ethnic minority, etc.) the researchers refrained from one-on-one interviews and instead opted for a focus group to allow a more relaxed peer-based discussion.

As a result of reaching theoretical saturation, we included responses from a total of 18 students. Semi-structured interviews in a focus group format were conducted. Such open-ended questions included whether students experience discrimination of any sort during their daily life, and if they have, detail the types of experiences and sources, as well as how that impacts their mental health. For participants who did not directly experience any discrimination, they were asked to describe what they have seen others experience instead. All responses were electronically recorded and transcribed verbatim, followed by de-identification of data per ethical board approval guidelines. Such qualitative responses were then thematically analyzed to identify emergent themes in a five-step process [31]. First, two independent researchers (C.M. and M.B.B.) read each focus group interview transcript at least two times and added initial concepts as marginal notes. Next, given that our focus was on discrimination, the initial coding utilized a theoretical thematic approach, versus inductive. Common words and phrases that were related to experiences of discrimination were highlighted. This was conducted independently by two researchers (C.M. and M.B.B.) and compared and contrasted until a consensus was reached. Once a list of identified codes were finalized, we reviewed each code and grouped into common themes, with consensus reached upon discussion. We next assessed each identified preliminary theme against Maguire and Brid Delahunt’s six-question checklist (do the themes make sense, does data support them, is there too much under a theme, are themes overlapping, and are there subthemes or other themes within the data. Finally, emergent themes were identified that were central to the qualitative feedback. During each step, supportive data (quotes) were selected to provide context of emergent themes.

## 3. Results

In our quantitative phase, a total of 308 participants were evaluated. As shown in Table 1, a majority of the study population were females (63%), aged 18–20 years (51.7%), and Hispanic/Latino (82.9%). Further, 48.4% reported low physical health status, defined as very poor/poor/average, 45.1% reported low mental health status (defined as very poor/poor/average), 21% reported serious psychological distress, and 37.4% were food insecure. Further, the mean everyday discrimination score of the entire study population was 2.10.

We then compared everyday discrimination score by stressors (low mental health, low physical health, serious psychological distress, and food insecurity) to evaluate the potential compounding role. Results show that mean everyday discrimination score was significantly higher among those with low physical health status, as compared to those with excellent/good physical health status (2.44 vs. 1.78, *p* < 0.01) and those with low mental health status versus those with excellent/good mental health status (2.40 vs. 1.87, *p* < 0.05). Likewise, such mean discrimination score was significantly higher among those with serious psychological distress, versus those without (2.97 vs. 1.89, *p* < 0.001). Further, participants who were food insecure reported a significantly higher mean discrimination score, when compared to their food secure counterparts (2.81 vs. 1.73, *p* < 0.001).

We also evaluated the commonly cited reasons participants felt they were discriminated against. As shown in Table 2, the most prevalent reason reported was appearance (57.9%), followed by race/ethnicity (46%), skin color (31.3%), and gender identity (21.8%).

Our qualitative results identified two central themes related to discrimination, including expectations, socio-demographic, and acculturative stress, as well as several emergent subthemes as participants described a plethora of sources and types of experiences of discrimination in their daily lives, which further provide insight into the association noted between discrimination and poor mental and physical health among participants. Interestingly, when assessing experiences of discrimination in daily life, a common theme was that of **norm** or **expectation**. For example, among Black/African American participants, the underlying notion was that as a person of color experiencing discrimination is routine.


*“…in a country that wasn’t set up for my people, you know? It was set up for us to fail.”*



*“Growing up being black you’re gonna, you’re gonna experience [discrimination].”*


On the other hand Hispanic/Latino participants noted more detailed encounters they were often surprised at the experiences based on stereotypes, aggregation, and generalization based on expected behaviors/mannerisms.


*“They judge us and see us as the same person… we may look the same, but we’re completely different.”*



*“My dad…likes lowrider cars and we’re like driving … we got pulled over, like we weren’t doing anything, but my dad got slammed like to the hood of the car… ‘cause they said he was gang affiliated, but my dad has never even been in a gang, but just because of like the way he looked”*



*“My boyfriend is white. He’s like why are you so Mexican? I could find someone better than you.”*


When assessing specific sources and types of discrimination, further theme of **socio-demographic-based discrimination** emerged. For example, **workplace discrimination** (including at sites of internship, volunteer work for professional development, etc.)**, primarily based on younger age,** as well as gender stereotypes were the most prominent theme among study participants.


*“… it sucks to know that someone is undermining a lot of the trainings that I went through and the experience that I got.”*



*“I think it’s because we are millennials. I’m guessing that’s what it is…that we’re not as competent compared to someone from another generation.”*



*“I wish I could be seen as a professional and not just as seen as …too young to teach them…”*



*“Just being a woman when it comes to like… heavy lifting, I kind of see guys tend to not like, pick me because I’m small. For example, I work at the gym, so it’s like lifting treads and all that it’s like let’s send the boys to do it.”*


Another common theme related to workplace discrimination was related to **elitism**, and based on educational attainment, discipline, as well as income. Participants noted that employees with higher rank often belittled others who did not hold a similar position or higher degrees. Responses show that choices in academic fields have also been discriminated against. Such experiences of discrimination that participants experienced further led them to quit their jobs and the associated stressors.


*“I work with a lot of doctors, nurses… they [have] higher education. I feel like sometimes I’m looked down upon.”*



*“Oh, well what are you doing? And I’m like oh, I’m doing public health and they’re like oh, well that’s not interesting.”*



*“…people with money they see all the other people a lot lower.”*


Furthermore, another emergent theme was that of acculturative stress, when the process of adapting to the norms of U.S. resulted in a negative impact on both domestic and international students. For example, most Hispanic/Latino and Black/African American participants noted experiences of **colorism from within the family**, where several participants noted that family members (including themselves) who were of darker complexion were often mistreated by others in the family of lighter complexion.


*“My Mexican side of the family will be like…You guys are black [referring to skin color and not race] anyways…I’ve kind of felt different in a sense… we’ve always been like kind of kicked to the curb.”*



*“I have two little cousins and they’re both the same age and one is like blond light-colored eyes and the other one is like darker, you know just a little bit tanner and he [grandfather] like willingly shows like he prefers like the lighter one.”*


Furthermore, participants also noted that within family, **language-based discrimination** occurred, where those not speaking a traditional language fluently often resulted in being called names or felt isolated. For instance, several participants described situations where having an American accent when speaking their family’s native tongue often resulted in being targeted and felt lack of belonging.

“They just started arguing with me like, okay, they were real… they’re like [I was] trying to be white or something.”

“…you don’t belong… when I try to speak Spanish… so that always kind of made me feel like a sense of loss of identity.”

Although experiences of discrimination on campus and/or classrooms were not widely reported, among those who did, such experiences were based on **being an immigrant**, and primarily due to having a foreign accent while speaking English or ethnic-specific stereotypes, which, in turn, participants noted left them with feelings of isolation or unwanted negative attention.


*“Foreign exchange students sometimes they get looked down upon… when they’re speaking.”*



*“I notice sometimes when I talk to people, they like to point out my accent.”*



*“Sometimes I feel a little different in class…because my English no good, no one wants to work with me.”*



*“One time during class we were choosing groups and I heard people say that [redacted ethnic identity] are lazy and only cheat.”*


Nearly all participants noted that the various forms of discrimination, especially related to work, negatively impacted their mental health. Many noted that such experiences, especially at work, made them question their own skills, abilities, worth, as well as added stressors with long-term feelings of hurt, isolation, and self-blaming.


*“I can’t do my job without having [the discrimination experiences] in the back of my mind.”*



*“…now I go to therapy because I talk about all of the stuff that I go through [at work].”*



*“When I go home… I relink what whatever happened… replay everything in my head.”*



*“It’s more like a frustration kind of thing… but it also makes me feel like maybe I’m not approaching them the right way… what am I doing that’s wrong?”*



*“Well, honestly I do get self-conscious about my accent, and I think about it all the time now.”*


## 4. Discussion

The purpose of our study was to assess experiences of discrimination, how that may influence mental health, as well as delineate the various types and sources of discriminations that such a population face. The results of our explanatory sequential mixed-methods study demonstrate that: experiences of discrimination were prevalent among college students with disproportionate shares among those who were food insecure, highlighting the most vulnerable. This is further clarified in qualitative analysis that highlights socioeconomic status-based discrimination, where participants’ education level and degree type at workplace was cited. Additional workplace-related discrimination included experiences based on younger age and gender identity. Furthermore, experiences of discrimination came from within the family based on colorism and having an American accent and from peers based on having a non-American accent, especially among immigrants. The expectation of discrimination, especially among Black/African Americans was also common. Cumulatively, various levels of discrimination (family, workplace, school, etc.) and types (language, work experience, education, etc.) demonstrate the complexity of the stressors college students experience that can negatively impact mental and physical health of the target population.

The literature notes that mothers (of young children) who experienced racial/ethnic discrimination were also more likely to report household food insecurity and poorer mental and physical health status [32], with similar patterns noted among 154 African American adults in South Carolina [33], and men of a sexual minority status [34]. Although similar studies among college students are limited, our study noted that not only are food insecure participants experiencing higher everyday discrimination, but such discrimination often occurs in areas where students are aiming to gain professional experiences (work, internships, volunteer work), which, in turn, negatively influences their self-efficacy in work performance. Given that food insecurity is known to lower academic performance [35,36], which, in turn, negatively impacts future employment [37], our results highlight that to optimize workforce development, employer training on discrimination practices based on socioeconomic status remains imperative.

Furthermore, our results highlight that a substantial ongoing theme of workplace discrimination based on both younger age, as well as woman/girl gender identity. Such results have significant implications for long-term professional development of college students. For instance, while historical legislation on ageism has focused on discrimination against those aged 40 years or older [38] and research on age-based discrimination among younger populations remains substantially negligible, there is a rising pattern in the empirical evidence that young adults face difficulties securing employment due to perception of lacking experience [23,39]. In our study we noted that even participants with substantial certifications and experiences often felt discriminated against due to either being younger in age or lacking higher education (masters or doctorate). This double-jeopardy of bias, resulting from societal perception that age or higher education alone equates to qualification, can posit a significant burden in the labor market, especially during the COVID-19 pandemic that is creating a global labor shortage [40,41]. In addition, participants further noted that their feminine gender identity was also associated with experiences of discrimination, independent of their qualifications; a pattern further noted in the literature as a common experience of women [42]. Cumulatively, the qualitative results of our study further clarify why a majority of participants in the quantitative assessment noted appearance as the most prevalent form of discrimination, as both looking younger, and gender were further reported as common workplace-related sources of discrimination. As such, legislation and policies that promote workplace training on qualification assessment beyond that of age or degree alone, and instead emphasizing competency, relevant experiences, etc., are needed. Likewise, promoting resilience among college students through career advancement workshops may provide the needed self-efficacy for recent graduates to self-advocate [23].

Furthermore, a unique theme noted in our study was family as a source of discrimination, especially based on colorism and having an American accent when speaking the family’s native language. Colorism results in advantages and privileges for those lighter skin, when compared to those of darker skin within racial/ethnic minorities [43]. Studies on colorism within the family, however, show inconsistent results. For example, when assessing discrimination based on skin tone/color among African Americans, results show that while families and media plays a critical role in developing and sustaining perspectives of colorism [44], colorism within family may either be limited or may not play a critical role in experiences of discrimination or mental health [45,46]. On the other hand, colorism has shown to play a role in employment among African American communities, with studies noting African Americans with light skin tones having higher ability to find employment [47]. Similarly, among the Hispanic/Latino population, the role of media in promoting colorism has been noted in the literature [48]. Our study, however, adds to this body of evidence by providing affirmation on the experiences of colorism that darker-skinned racial/ethnic minorities face within their families and resulting in feelings of not belonging and isolation.

In addition, our results provide insight into the role of accents and the catch-22 that racial/ethnic minority college students face. For instance, while having an American accent while speaking a native language within one’s family was related to experiences of discrimination, isolation, and sense of not-belonging, having a non-American accent, on the other hand, was a source of accent-based discrimination among peers. Although limited, the literature notes the prevalence of accent-based discrimination among international students and the negative burden of mental health [49,50], as well as interracial othering among Asian Americans [51]. Our results demonstrate that the type of accent-based discrimination differs among racial/ethnic college students depending on the audience, with contradictory stressors from family versus peers. Although alleviating accent-based discrimination is complex, involves historical and cultural basis, as well as normalization of mocking non-American accents in media [52,53], there remains a critical need to address such linguistic racism [52] as an integral part of anti-racist movement. Campus-based initiatives that promote course content, such as video lectures, from linguistically diverse professionals may provide a simple and yet effective pedagogical initiative to create more inclusive practices in the classroom. Likewise, resiliency building among students to address experiences of discrimination within a family and the ability to defer from maladaptive coping mechanisms may be beneficial.

Finally, a concerning pattern noted in our qualitative assessment was the expectation of racism, especially for Black/African American college students. Herein lies opportunities for institutes of higher education to be active in their diversity, equity, and inclusivity initiatives through integration of culturally responsive and social justice pedagogy [54,55], in turn promoting the next generation of anti-discrimination advocates.

The results of the study should be interpreted in the context of its limitations and strengths. Although our study utilized a mixed-methods approach, the cross-sectional nature limits our ability to assess the long-term impact of experiences of discrimination among college-students. Additional dimensions of discrimination were not assessed in the quantitative assessment, especially those stemming from acculturative stress for immigrants, sexual orientation, or others. Nevertheless, our qualitative assessment provided means to lower this bias by providing additional content on experiences of socio-economic and linguistic discriminations experienced by the target population.

Cumulatively, our study highlights that college students experience a plethora of discrimination from family, peers, as well as during opportunities of professional development. Optimizing mental health outcomes and overall well-being of the population requires both public health and legislative initiatives that promote the creation of an inclusive environment and resilience building.

## 5. Conclusions

Experiences of discrimination are prevalent in the U.S., especially among vulnerable populations. Most of studies related to discrimination, however, focus on racial/ethnic and sometimes gender-based experiences. In our study, we aimed to evaluate both the unique types and sources of discrimination among college students as they often face a plethora of different environments during this transition period. Our results are consistent with the literature in highlighting that discrimination is prevalent among college students. However, such findings further note that such experiences are significantly related to their health outcomes, including mental health and perceived self-worth. In particular, the results note that experiences of discrimination may benefit from assessing social class in America, acculturation-related patterns, as well as whether expecting such stressors can lead to maladaptive coping mechanisms. Workplace training as well as campus-based initiatives are needed to promote inclusive environment for college students.

## Figures and Tables

**Table 1 ijerph-19-09607-t001:** Study population characteristics (n = 308).

Sex	
Female	63.0%
Male	37.0%
Age (years)	
18–20	51.7%
21–23	33.8%
24 or older	14.6%
Ethnicity	
Hispanic/Latino	82.9%
Not Hispanic/Latino	17.1%
Food Security Status	
Food secure	62.6%
Food insecure	37.4%
General Physical Health Status	
Excellent/very good/good	51.6%
Very poor/poor/average	48.4%
General Mental Health Status	
Excellent/very good/good	54.9%
Very poor/poor/average	45.1%
Psychological distress	
Serious psychological distress	21.0%
Not serious psychological distress	79.0%
Mean everyday discrimination score (standard error)	2.10 (0.12)

**Table 2 ijerph-19-09607-t002:** Cited reasons for discrimination among study population.

Appearance	57.9%
Race/ethnicity	46.0%
Skin color	31.3%
Gender identity	21.8%
Religion	12.3%
Sexual orientation	7.6%
Immigration status	5.2%
Disability	3.8%
Other	5.2%

## Data Availability

Data are not available per IRB guidelines on dissemination.

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
