# Peer review of "“We’ve Always Been Kind of Kicked to the Curb”: A Mixed-Methods Assessment of Discrimination Experiences among College Students"

_ijerph, 2022, doi:10.3390/ijerph19159607_

Round 1
Reviewer 1 Report
The paper addresses a very important topic that needs ongoing scientific analysis: the relation of discrimination and (mental) health. The empirical study collected interesting data on this phenomenon. Nevertheless, I suggest a thorough revision, especially regarding the theoretical framing of the phenomenon of discrimination.
1) Background
Could you phrase the research interest respectively the concrete research questions more directly/precisely. Please clarify as well how they are deducted from the described background.
Could you give a theoretical framing of your understanding of discrimination? In which theoretical debate is this understanding located? What do you subsume under the term discrimination in this paper?
Line 43: I am not sure what is meant by „economic behaviors“. It would be helpful for the reader to clarify that.
Line 54/55: From my point of view the conjunction of health disparities in general and health disparities in consequence of discrimination could be elaborated more.
Line 67/68: Here the “unique characteristics of college students” are mentioned. It would be helpful to clarify this uniqueness. In doing so, the specific focus on this group should also be more clearly elaborated and justified
Line 64ff: What is meant by “limited to assessment of experiences of discrimination based on racial/ethnic identity and/or colorism and sometimes gender norms”? Should other forms of discrimination be included? Which and why?
2) Methods
Overall: How do the different methods tackle different aspects of the research interest?
Line 73: As it is stated that the study is explanatory, it would be important to have knowledge about the concrete hypotheses that were tested.
Line 106: Why was the qualitative data collected via focus groups (collectively)? What was the benefit of this method?
Line 113: Please give a little bit more information about the qualitative data analysis (e.g. the methodical literature you refer to).
3) Results
The qualitative analysis focuses on very different types/sources/topics of discrimination. It would be important to differentiate more clearly how they are interconnected, on which level the different types/sources/topics are located?
Line 133: What does the category “appearance” include? How is it defined in contrast to other categories (e.g. skin color)?
As you stated in line 89 you dichotomized the race/ethnicity variable to Hispanic/Latino vs. not Hispanic. I am not quite sure how the experience of Black/African Americans (line 140ff) is connected to that dichotomization. Could you clarify how you dealt with these different ‘categorizations’ on the different levels of analysis?
The connection between experiences of discrimination and health aspects would still have to be specified, also with regard to different forms of discrimination. A more in-depth analysis of the qualitative data could also contribute to this (e.g. a closer look at the difference between different forms of discrimination, e.g. difference between open violence vs. questions of recognition of one's own performance).
4) Conclusion
The conclusion could be more specific, especially in regard to existing literature on discrimination and (mental) health. References to the theoretical background – to be elaborated in the introduction – would be helpful for the categorization of the outcome.
Author Response
Response:
Thank you for the detailed review. We have addressed all comments.
We have updated the background section to define discrimination, add more literature on the outcomes, clarify economic behaviors, and why college students are considered unique (based on literature), as well as the different types of discriminations that may exist beyond the norm race/ethnicity.
Per the feedback, we have noted the research aims. We have also added how each phase is unique, it’s purpose, and expanded the data analysis of the qualitative section.
We appreciate the guidance on presenting the results and further expanded. We also noted that independent of race or ethnicity, colorism was a consistent factor among Hispanic/Latino and Black/African American college students. Given that appearance can include more than skin color, and may address factors such as body weight, scars, etc. we included the broader option, in addition to skin color. We have updated the conclusion as well.
Reviewer 2 Report
The article refers to an interesting investigation in which the impact of discrimination in several dimensions on the mental health of young university students from an ethnic minority (Hispanic) is crossed, but the author’s needs to improve some aspects:
In order to improve the presentation I suggest that:
- In the summary, it should be explained what the qualitative methodology consisted of.
- In terms of contextualization, it would be necessary to insert some information about the different ethnic groups in the USA and the reproduction of stereotypes about certain populations. Sometimes, the argument appears "oriented" in the sense that there will be some direct relationship between the situation of discrimination and food shortages and poverty. I think there is a risk of reproducing stereotypes that exist about some population groups.
- In methodological terms, I understand that a better description of the participants is required: what are the selection criteria? What are the social and economic characteristics of the parents? Who are the participants in the quantitative study and in the quantitative study? This is not clarified, everything appears a little as a homogeneous mass.
- In terms of the presentation of results, I think it would be beneficial to try to assess whether there are differences in results according to the social and family backgrounds of the participants, perhaps with the construct of social profiles. In the presentation of the testimonies of the participants, it will be necessary to identify who says what: at least gender, age and education. I think that the profession of parents and schooling could also help us to build social portraits and to think in different cases, not considering all as the same. It is important to give a global overview of the different situations encountered.
Author Response
Thank you for the detailed review. We have addressed all comments as noted below.
We have added the qualitative method description in the summary (abstract).
We have added ethnic group definitions of the U.S. in the methods.
We have added selection criteria and other available content. Given all participants are of legal age in the U.S., parental information is not available, but we have added overall family information when available for the population of recruitment. We have also delineated, when feasible, the demographics of the qualitative and quantitative participants.
Given that the participants are of legal age in the U.S., we could not collect family data, however, we do provide population data in terms of socioeconomic status. Among participants we also used food security, as a marker for poverty, given the high rates of poverty in the area as well. Due to the means by which our review board was approved, we were not allowed to collect demographics of qualitative participants, other than ensuring they matched quantitative, which is further clarified in the methods. The content on global overview is provided in the scope of social stressors, but again, due to legal age of students, who thus do not live with family, we cannot assess profession or parents’ education. However, we do note in our methods that majority are first generation college students on financial aid.
Round 2
Reviewer 1 Report
Thank you for the revised version and for adressing the suggestions of the first review (especially in the introduction & the method section). The background and procedures are much clearer now! Nevertheless, in the result section and the conclusion I still suggest some revisions.
The results seem to be located on very different levels (as mentioned in my first review). Can you make clearer why the different 'themes' condensed from the empirical are on such different levels, how are they neverthelesse interconnected?
187ff As a first theme you mention the aspect of espectation. As I understood it, you found a difference between Black/African American (expected discrimination) and Hispanic/Latinos (are surprised by d.). The examples you give below seem not in line with this differentiation? Can you clarify that?
203ff What is meant by socio-demographic based discrimination here? In the following you write about workplace discrimination. From my point of view that are two different levels; the first is a source/reason (a very general one) of/for d., the second a context, in which d. takes place. Can you clarify that?
The conclusion could be still more specific, especially in regard to existing literature on discrimination and (mental) health.
I hope the suggestions for for revising the paper are helpful for you!
Author Response
Thank you for the revised version and for adressing the suggestions of the first review (especially in the introduction & the method section). The background and procedures are much clearer now! Nevertheless, in the result section and the conclusion I still suggest some revisions.
The results seem to be located on very different levels (as mentioned in my first review). Can you make clearer why the different 'themes' condensed from the empirical are on such different levels, how are they neverthelesse interconnected?
Thank you for the feedback. Given these were analyzed themes from the qualitative, we reported the emergent ones. However, in the discussion we have added further interpretation as to why the different types have occurred and their putative interconnectedness.
187ff As a first theme you mention the aspect of espectation. As I understood it, you found a difference between Black/African American (expected discrimination) and Hispanic/Latinos (are surprised by d.). The examples you give below seem not in line with this differentiation? Can you clarify that?
Thank you for the feedback. We have moved the quotes immediately after each section to help see the connection better.
203ff What is meant by socio-demographic based discrimination here? In the following you write about workplace discrimination. From my point of view that are two different levels; the first is a source/reason (a very general one) of/for d., the second a context, in which d. takes place. Can you clarify that?
Thank you for the feedback. We have clarified the mention of education level from workplace to provide better context.
The conclusion could be still more specific, especially in regard to existing literature on discrimination and (mental) health.
We have updated the conclusion. Thank you for your feedback.
I hope the suggestions for for revising the paper are helpful for you!